# Qualitative systems mapping for complex public health problems: A practical guide

Anneleen Kiekens[1]*, Bernadette Dierckx de Casterlé[2], Anne-Mieke Vandamme[1,3]

**1** Department of Microbiology, Immunology and Transplantation, Rega Institute for Medical Research, Clinical and Epidemiological Virology, Institute for the Future, KU Leuven, Leuven, Belgium, **2** Department of Public Health and Primary Care, Academic Centre for Nursing and Midwifery, KU Leuven, Leuven, Belgium, **3** Center for Global Health and Tropical Medicine, Unidade de Microbiologia, Instituto de Higiene e Medicina Tropical, Universidade Nova de Lisboa, Lisbon, Portugal

* Anneleen.kiekens@kuleuven.be

## Abstract

Systems mapping methods are increasingly used to study complex public health issues. Visualizing the causal relationships within a complex adaptive system allows for more than developing a holistic and multi-perspective overview of the situation. It is also a way of understanding the emergent, self-organizing dynamics of a system and how they can be influenced. This article describes a concrete approach for developing and analysing a systems map of a complex public health issue drawing on well-accepted methods from the field of social science while incorporating the principles of systems thinking and transdisciplinarity. Using our case study on HIV drug resistance in sub-Saharan Africa as an example, this article provides a practical guideline on how to map a public health problem as a complex adaptive system in order to uncover the drivers, feedback-loops and other dynamics behind the problem. Qualitative systems mapping can help researchers and policy makers to gain deeper insights in the root causes of the problem and identify complexity-informed intervention points.

## Introduction

In recent years, systems thinking methodology is increasingly used to study health systems and complex public health problems [1, 2]. Researchers and policy makers around the globe are more and more aware of the need to shift away from reductionist cause-effect approaches towards a systemic understanding of public health issues [3]. Health systems may be conceptualised as complex adaptive systems (CASs), which entail a set of diverse, interrelated factors and which are characterized by dynamic behaviours such as emergence, self-organization and the formation of feedback-loops [4–6]. In a CAS, positive interventions in one part of a system, may have undesired effects in other parts of the system, depending on the paths that exist in the system and choices and events that happen along the way. This phenomenon is called path dependence.

Despite the rising interest in systems approaches, literature on the topic remains dispersed and a common jargon is yet to be developed [1]. Moreover, the available literature on complex

**Data Availability Statement:** All relevant data are within the manuscript and its Supporting Information files.

**Funding:** The author(s) received no specific funding for this work.

**Competing interests:** I have read the journal's policy and the authors of this manuscript have the following competing interests: AV declares consultancy fees from Gilead. This does not alter our adherence to PLOS ONE policies on sharing data and materials.

systems approaches in the field of public health has remained largely theoretical. A commonly used method to visualise, understand and analyse a CAS is systems mapping. Systems mapping has been used to study diverse public health problems such as obesity, vaccine hesitancy and neglected tropical diseases [7–9]. The term systems mapping comprises a set of different methods for visualising and analysing complex adaptive systems. Depending on the exact nature of the research question, a different type, or combinations of types, of mapping can be used. One of the most used types of systems mapping is causal loop diagramming [1]. This is a qualitative approach in which the causal relationships between factors are represented. Connections between elements are directed and can be positive (both elements evolve in the same direction) or negative (both elements evolve in the opposite direction). An often-used example of a causal loop diagram is that of the heating and the thermostat (Fig 1). When the room temperature drops to a certain point, the thermostat will automatically increase the heating (negative causality). When the heating is on, the room temperature will increase (positive causality) up until the point in which the desired temperature is reached.

While in this manuscript we primarily focus on the mapping of causal loop diagrams, there are also other types of mapping such as stock and flow diagrams, which is a more quantitative approach to systems mapping and can be used to study the dynamic behaviour of a system over time, or social network analysis which aims at visualising and studying the relationships between social actors [10, 11]. Such visualisations of CASs help to gain deeper insights into the dynamics of complex problems and to develop a shared understanding between different stakeholders in order to come to a nuanced understanding of the complexity of the situation. Systems dynamics and types of modelling have been thoroughly described by Sterman (2000) [12]. Systems mapping aims to do more than integrating the perspectives of different stakeholders. It uncovers emerging dynamics which are built up of more than the sum of the elements involved and which would likely have remained uncovered if a linear approach were adopted. In some cases, the process of developing a systems map may be more valuable than the final product itself. In such cases, participatory practices such as group model building sessions may be used. Different stakeholders then come together to jointly develop insights and search for solutions while mapping the system [13–15]. However, group model building sessions are not always the most desirable or feasible option. For example, the topic under investigation might be highly stigmatised in the community, therefore not allowing participants to speak freely during a group model building session, the participants perspectives on the topic may be too diverse to organize a common discussion (e.g. technical experts vs family and friends), or participants might live in different parts of the world, making a physical meeting organizationally challenging [16, 17]. In some cases, individual interviews are also preferred when one aims to understand the individual mental models of stakeholders separately before generating an integrated overview of the combined viewpoints [18]. In this article we provide some practical guidelines and reflections on how to use systems mapping as a means to collect and investigate rich, complex data on public health challenges in order to integrate different perspectives and gain a deeper understanding of the complex systems dynamics. We use a case study concerning HIV drug resistance in sub-Saharan Africa as a practical example to clarify how complex data can systematically be collected, mapped and analysed while incorporating the principles of transdisciplinarity every step of the way (Fig 2) (Table 1).

## Case study

The methodology discussed in this paper is illustrated by a study on the complex adaptive system of factors influencing HIV drug resistance (HIVDR) in sub-Saharan Africa [19, 20]. Although antiretroviral therapy (ART) is available, allowing people living with HIV to live a

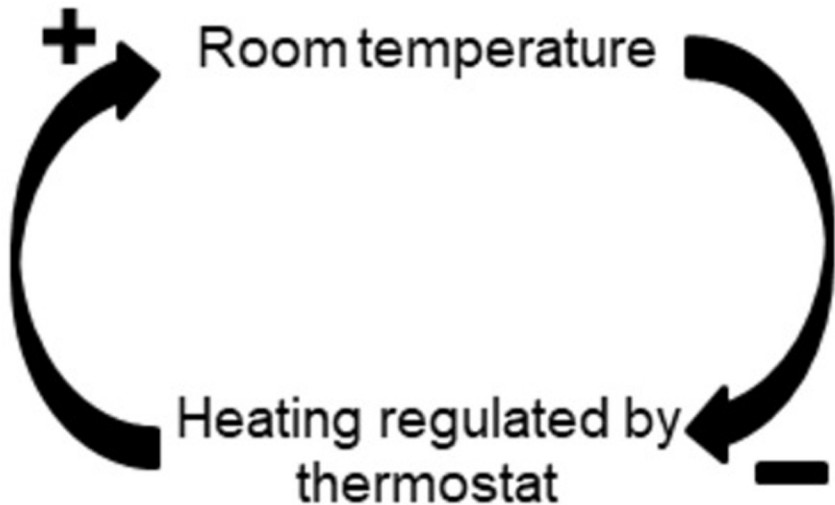

**Fig 1. Causal loop diagram example.** Thermostat room temperature regulation as a (simplified) example of a causal loop diagram.

long and healthy life, the increase in HIVDR is threatening the success of the available therapies. HIVDR arises when the ART present in the body is insufficient to suppress the viral load, creating selective pressure which allows the virus to mutate in order to escape the effect of the therapy. This situation can be due to irregular adherence of PLHIV to their therapy which on its turn has many other possible causes. The aim of the study was to gain detailed insights in the underlying dynamics of factors influencing HIVDR and to identify suitable intervention

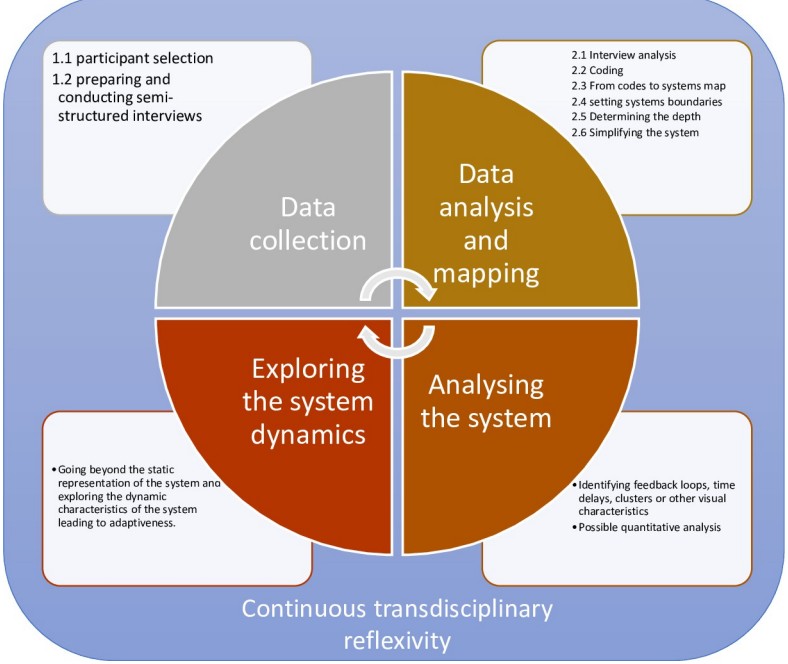

**Fig 2. Graphical abstract.** Overview of the described methodology, consisting of four iterative building blocks and continuously requiring the researchers to adopt a transdisciplinary approach and to be aware of their disciplinary biases.

**Table 1. Overview of the guideline and the timing of each step throughout the process.**

| Step | Description | Timing |
|------|-------------|--------|
| 1 | **Data collection** | |
| 1.1 | Participant selection | |
| 1.2 | Preparing and conducting semi-structured interviews | May lead back to 1.1 |
| 2 | **Data analysis and mapping** | |
| 2.1 | Interview analysis | May lead back to 1.1 and 1.2 |
| 2.2 | Coding | May be in parallel with 2.1 |
| 2.3 | From codes to systems map | After 2.2 |
| 2.4 | Setting systems boundaries | After 2.3, though one can reflect about this throughout the data analysis |
| 2.5 | Determining the depth of the system | After 2.4, though one can reflect about this throughout the data analysis |
| 2.6 | Simplifying the system | After 2.5 |
| 3 | **Analysing the system** | In parallel with 4 |
| 4 | **Exploring the system dynamics** | In parallel with 3 |
| 5 | **Continuous transdisciplinary reflexivity** | Throughout the whole process |

points. To this purpose two systems maps were developed: one visualising the complex adaptive system of factors influencing HIVDR as understood by experts on an international level and one visualising the system at local level for a study site in Dar es Salaam in Tanzania. These systems maps were informed by interviews with 15 international experts, 12 PLHIV in Dar es Salaam and 10 local actors, who through their daily activities regularly come into contact with PLHIV in the study site. The findings of these studies are described in two separate publications [19, 20]. In the next sections we use examples from this case study to illustrate our guide.

## Guidelines

**1. Data collection.** *1.1. Choosing a data collection method and participant selection.* The first consideration to make is which way of collecting data is most suitable for the topic under investigation. As already explained in the introduction, there are different reasons (both methodological and practical) to opt for either group modelling sessions or individual interviews. This guide focusses specifically on the mapping and analysing of complex data collected by semi-structured interviews. Participants should be recruited with the aim of obtaining a full picture of all aspects of the system. Next to interviewing patients and healthcare workers, one might therefore also consider interviewing people who are somewhat further removed from the core problem but are still in touch with certain parts of it. For example, architects designing certain hospital area's relevant for the topic under investigation or religious leaders who provide spiritual support to patients could contribute unique insights into the topic. Next to the interviews themselves, other types of data such as participant observation and document analysis could also be used to triangulate the data and increase the validity of the results.

In our case study we opted for collecting the data for our systems map through individual semi-structured interviews. This had a dual reason. First, for our mapping at international level, we wanted to obtain broad insights in all possible factors influencing HIVDR. Individual interviews were chosen to give us the opportunity to collect deep insights in the specific expertise of the interviewee while verifying or building further on information obtained from previous interviews. For our local map, individual interviews were preferred for another reason. As

people living with HIV still face strong stigmatisation, we wanted to create a safe environment for them to speak their mind, without other community members present. Moreover, a workshop would likely only have attracted PLHIV who felt comfortable with their HIV status and were facing less difficulties adhering to the therapy, while for the interviews we also managed to recruit PLHIV who had dropped out of care.

*1.2. Preparing and conducting semi-structured interviews*. A semi-structured interview guide should be developed based on the available scientific literature or already existing and validated guides on the topic, and adapted throughout the data collection process when new insights are developed. In order not to bias the data collection towards certain assumptions the researcher may have, it is advised to start the interview with a broad, open question about the complex problem, rather than asking a question about a single aspect of the problem. This way, the participants are inclined to start by expressing the aspect of the complex problem most important to them and the interviewer can explore the main believes and experiences the participant has to share about the topic. For example, an initial question like "how do you experience being HIV positive" may reveal to the interviewer that the patient's whole perception of his or her HIV infection is based on the believe that it is a punishment of God. This information is important for the interpretation of the rest of the interview and may not have come up if the interview had started with a focus on a certain aspect of the system, such as the question "how do you perceive the healthcare service you receive?". After this first question, the interviewer may continue covering a list of specific topics, retrieved from the literature or which came up in previous interviews. When a question is answered by "A happens because of B" the interviewer can ask for specific examples or experiences that support this claim and subsequently delve deeper into other possible underlying causes aiming to obtain the structure "A happens because of B, which is caused by C, D and E, etc." This continues until a sufficient level of depth is reached or until the insights of the interviewee are exhausted, at which point the chain of causality may be built up further during interviews with other participants. Such chain of causality is built several times within one interview, each time starting from an open question. To further reduce bias, the interview circumstances should be well thought-through in order to create trust between the interviewer and interviewee. For example, interviews with PLHIV are best done in their native language and in a location that cannot be perceived as stigmatizing.

**2. Data analysis and mapping.** When developing a systems map based on the interview data, the first steps are largely similar to conventional interview analysis methods. The first steps of our data analysis method are inspired by the Qualitative Analysis Guide of Leuven (QUAGOL) which provides clear steps for capturing the rich insights in complex qualitative data [21, 22].

*2.1. Interview analysis*. Interviews should be recorded and transcribed verbatim. Throughout the interview process, the research team should have regular debriefing sessions to allow for modifications of the interview guide if needed. This can for example be the case when a new relevant topic comes up, which needs to be further investigated in the following interviews. Ideally, technical reports are written after each interview, describing the context of the interview, possible technical issues, characteristics of the participant and possible cultural clarifications important for the full comprehension of the data in their specific context. After thoroughly reading the transcripts, a series of meetings is organized between the research team in order to discuss the interpretation of the interviews and to make sure cultural elements are well-understood. It is advised to include researchers from different disciplinary backgrounds in the team, in order to prevent disciplinary bias in the analysis of the data.

*2.2 Coding*. Once an interview and its core messages are well understood, the coding process can start. Coding can be done with professional programmes such as NVivo or in an excel

table. The researcher keeps a list of all elements that were mentioned as a direct or indirect cause of the problem under investigation and of each link between two of those elements. For further analysis purposes, other types of data can be stored behind each element or connection. In Table 2 we explain the types of data that can be stored behind one element, using the element "accessibility of healthcare centre" as an example.

*2.3. From codes to systems map.* While keeping the codebook updated after analyzing each interview, it can also be of interest to make a separate systems map of each transcript, visualizing the mental model of the interviewee in order to understand how he or she perceives the system (Fig 3). Mental models are graphical representations of how people internally understand causal relationships between elements to make sense of a complex problem [23, 24]. They often unconsciously affect our behavior or decision making and are useful for the researcher to gain a deeper understanding of the interviewees way of thinking about the problem [25].

In a next phase these schemes can be merged manually or automatically by simply uploading the codebook in the used mapping tool. In our case study we used KUMU, a user-friendly online mapping tool which allows the storage of different types of data behind each element

**Table 2. Coding examples.**

| Data type | Explanation | *Example* | Note |
|---|---|---|---|
| **Element Name** | The factor directly or indirectly influencing the problem under investigation. | *Accessibility of healthcare centre* | |
| **Definition** | It is important to define the element and what is included or excluded in order to facilitate the interview coding. | *Accessibility refers to road access, public transportation, road safety, transport costs, limited opening time, poor access due to other disease outbreaks or wars etc. Distance to healthcare centre is considered a separate element.* | |
| **Number of Interviews** | The number of interviews a certain element or connection was mentioned in. | *7* | This element has been discussed in 7 out of 22 interviews. |
| **Interview Identification Number** | The identification number of the interviews in which a certain element or connection was discussed. | *For example*: I01, I03, I04, I08, I09, I13, I15 | (Fictive identification numbers are used due to confidentiality reasons). |
| **Quote** | The interview quotes in which the element or connection was described. Storing this information in the systems map will facilitate the analysis as all the quotes linked to a certain element can easily be revisited. | *"Sometimes I don't get a bus fare but I borrow somewhere because I must go for refill. When few drugs for two or three days remain, that is when I go to refill my drugs. I must go the same date written on my card by the health care providers so as I may not confuse them. If it is written fifth I must go to refill, so even if it is from my neighbour I borrow one thousand shillings so as I go to the facility to refill my drugs."* | This is one quote given as an example. During data collection, all quotes relevant to this element would be collected here. |
| **Tag** | The opportunity to categorise elements. | Healthcare system related | This allows the researcher to easily filter out all elements related to a certain topic, in this case healthcare system related factors. |
| **Other. . .** | Several other types of data (for example: degree of importance) can be stored, depending on what may be useful during the analysis process. | | For example, a degree of importance as judged by the interviewee could be given to elements based on how the elements or connections were described in the interviews. However, as this is a subjective indicator, it is advised to always use this parameter in combination with other ones when drawing conclusions. |

Examples of types of data which could be retrieved during the coding process and stored behind elements and connections of the systems map. We illustrate with an example of our study on HIVDR.

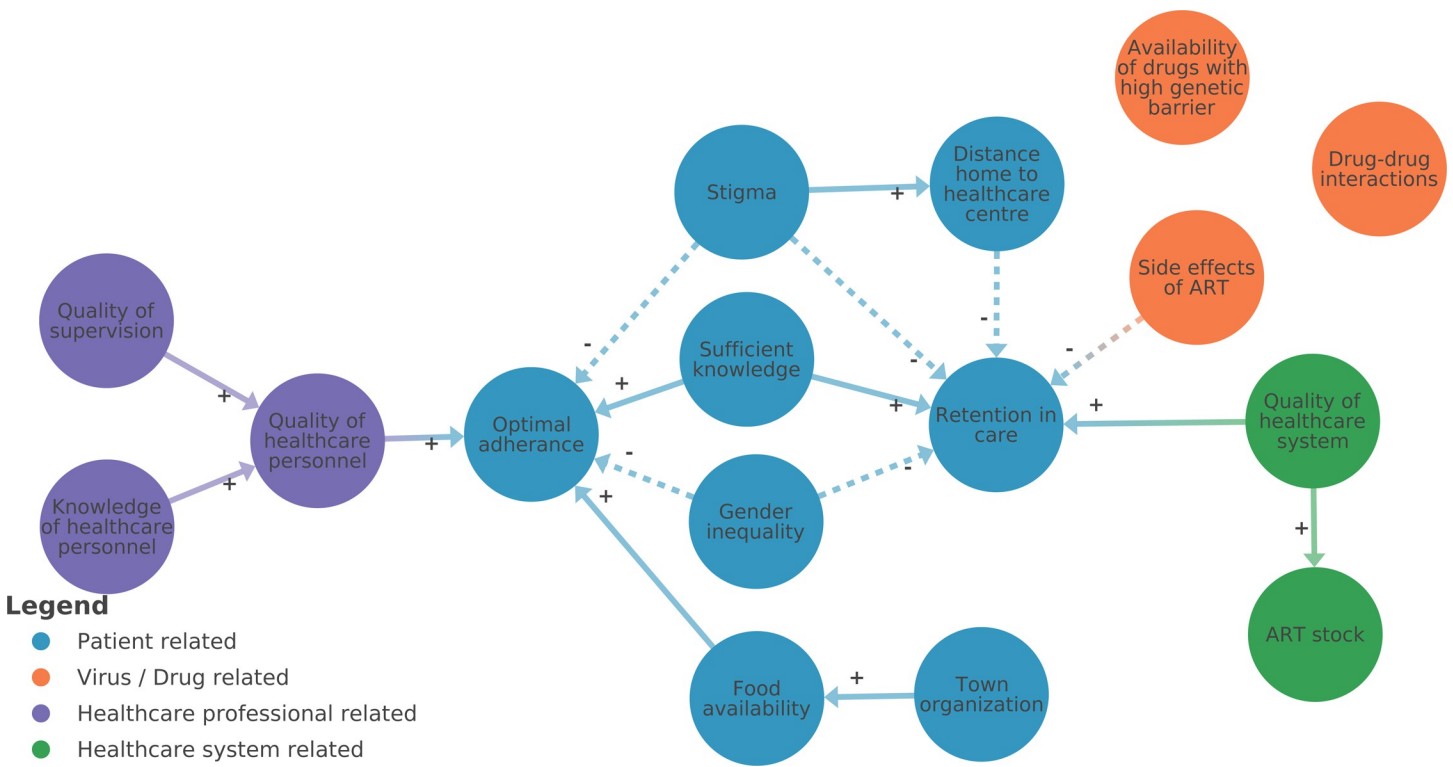

**Fig 3. Mental model example.** Example of a mental model of an interviewee, visualizing the elements and connections which came up during the interview and which are perceived to be true by the interviewee. The researcher tried to bring some first structure in the model by using a color code.

and connection and which has some built in analysis tools [26]. Once the codebook is imported, some immediate structure can be brought into the map by for example coloring or grouping the elements according to a common parameter or sizing the elements by number of occurrences. This structure will most likely be changed at a later stage in the analysis process when new insights are gained. Eventually, the researchers may opt to retain different visualizations of the model to highlight different structures. In the case of HIV drug resistance in sub-Saharan Africa, we developed one visualization showing how the elements relate to different societal layers, whereas the second visualization highlights the main dynamics of the system (Fig 4). In both visualizations each element is a factor influencing HIVDR as mentioned in the interviews and each connection indicates the relationship between those factors. An overview of all elements and connections is included in these maps as well as an interactive version of the systems maps where the reader can zoom in and click on elements and connections for more information is included in the supplementary files (S4 File) [27].

When a first basic structure is reached, we suggest to revise all elements and connections in order to avoid the same concept being visualized in different ways inside the map. For example, a pathway representing the difficulties PLHIV may face reaching the clinic due to their economic status may be represented as "economic status -> retention in care" or as "economic status -> ability to pay transportation -> retention in care". Especially when the coding is done by more than one researcher, the codebook may contain such double pathways. To resolve this, the research team has to come to a common agreement on how to visualize such concepts.

*2.4. Setting systems boundaries.* Throughout this process the researcher can also start to set system boundaries, determine the level of depth the CAS will be represented in and simplify

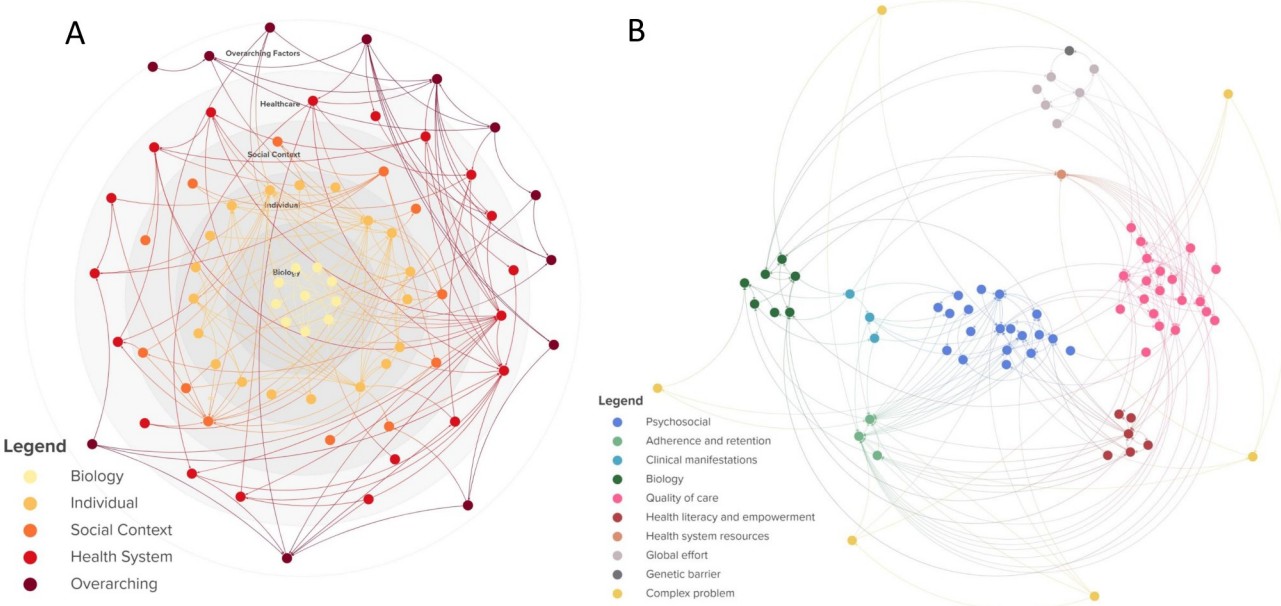

**Fig 4. Different ways of visualizing a system.** The elements and connections in A and B are exactly the same. In A the system is organized according to the different layers ranging from biology on the micro level to the individual level, the social context, the healthcare system and overarching factors at the macro level. In B, the elements are divided in thematic clusters and the relationships between clusters are visualized. Figure adapted from Kiekens et al. [19] and for illustrative purposes only.

the system. In reality, the boundaries of CASs are often blurry as different CASs are interlinked and systems are constantly evolving [4]. For example, while our case study aimed at covering a public health issue (HIVDR), we realized throughout the study that our system is strongly interlinked with other complex systems such as poverty (e.g. having financial means to reach the healthcare center and food insecurity influencing adherence as medication needs to be taken with a meal). However, when visualizing a system, some choice in what to include in the system and what not, needs to be made. Though it might seem tempting to set boundaries at the beginning of the project, the authors recommend starting without boundaries in mind and representing the CAS as detailed as possible. While more time consuming, the advantage is that the possibility of excluding important factors due to preset limits is reduced and the researchers gain deep insights in all aspects of the system before analyzing it or reducing it to its essence if needed. Boundaries can be set in different ways depending on the information the map needs to transmit. For example, one can decide to consider the factors which are not part of a closed system as being exogenous, meaning they only have influence on the system but are not influenced by the system. For example, in Fig 4B, all endogenous elements are part of a closed feedback loop (they influence and are influenced by the system), whereas the exogenous factors (indicated in yellow), are exerting an influence on the system but are not influenced by the system (e.g. "gender inequality" influences "HIV status disclosure" and "HIV transmission" but is not influenced by any element in the system). Another way of determining the boundaries of the system could be to view all elements that form the core of a different CAS as exogenous factors (for example: gender inequality, poverty and war and disease outbreaks are all complex problems on their own, which are interlinked with our complex problem).

*2.5. Determining the depth of the system.* The depth of the system refers to the level of detail a system is represented in. Issues surrounding stigmatization of PLHIV could be separately

represented as "stigmatization", "self-stigmatization", "gossip and discrimination" or as one common term such as "Stigma and discrimination" (Fig 5). Again, this depends on the research question and purpose of the systems map.

*2.6. Simplifying the system.* Additionally, the systems map might need to be reduced or simplified to a smaller, more manageable system that is understandable for external stakeholders. In the rest of this paragraph we suggest some strategies for the simplification of systems maps. Other strategies (possibly topic dependent) could also be used. More important is to consequently apply the strategy to the whole systems map. When in doubt whether two elements should be merged or not, we suggest the researcher asks two questions: 1) are there significant differences in nuance between the content of both elements? And 2) do both elements have different connections to other elements? If the answer to both questions is "No", the elements can be merged into one. Moreover, elements that have only one incoming and one outgoing connection (A->B->C) might be deleted and taken up into one connection from A to C (A->C), unless element B is crucial for the understanding of the system. When several loops are present, loops sharing a same broader theme can be summarized into one. This can be compared with a route on a roadmap [19]. When one wants to know the route from Paris to Brussels, there are several options. All the options pass by different towns but they all have one common theme: they represent ways to go from Paris to Brussels. Bundling these loops or pathways between two elements, may help to drastically simplify the map and to visualize only the core essence. Fig 6 is an example of a holistic, detailed system (A), summarized into its core feedback loops (B).

**3. Analysing the system.** In fact, the analysis of the CAS starts during the mapping process itself. Throughout the mapping process, the researcher will start to identify certain characteristics of the system. These could be for example reinforcing or balancing loops, time delays or clusters of elements or connections sharing the same characteristics. Though the analysis is foremost qualitative, involving a continuous (re)-reading of interview quotes or relevant literature, some quantitative elements may support the interpretation. McGlashan et al. propose some quantitative network analysis metrics and describe how to interpret them when applied to systems maps [28]. The in-degree describes the number of incoming connections (the number of elements influencing the element of interest). The higher the in-degree, the more the element is directly influenced by other elements of the system. In our case study, the element with the

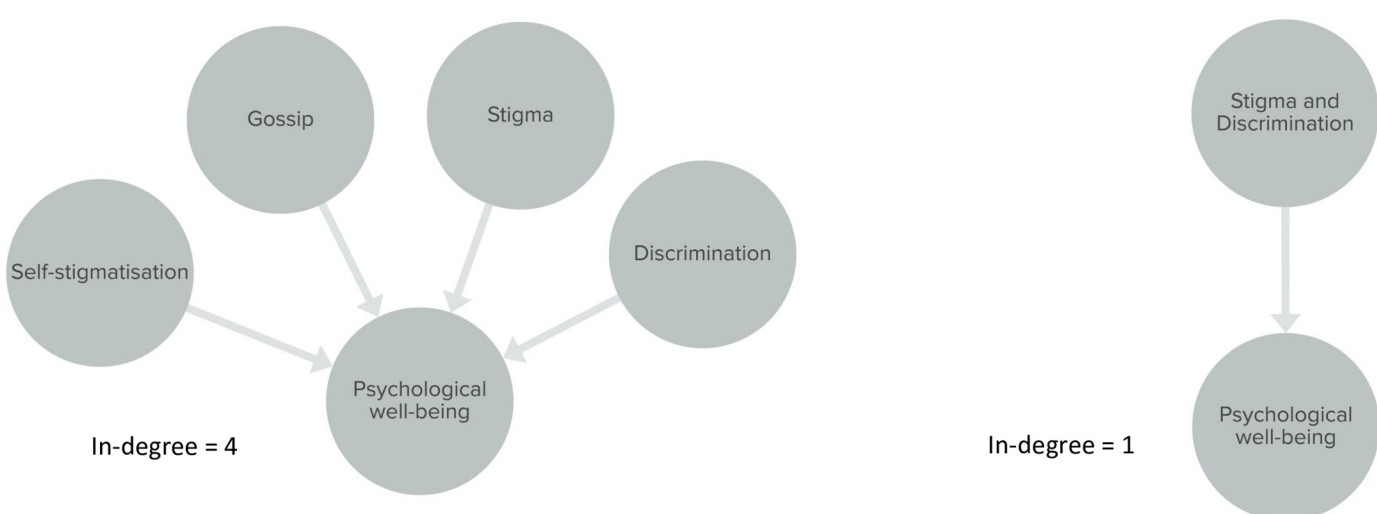

**Fig 5. In-degree.** Illustration of mapping choices to be made by the researchers and the consequences for the in-degree metric.

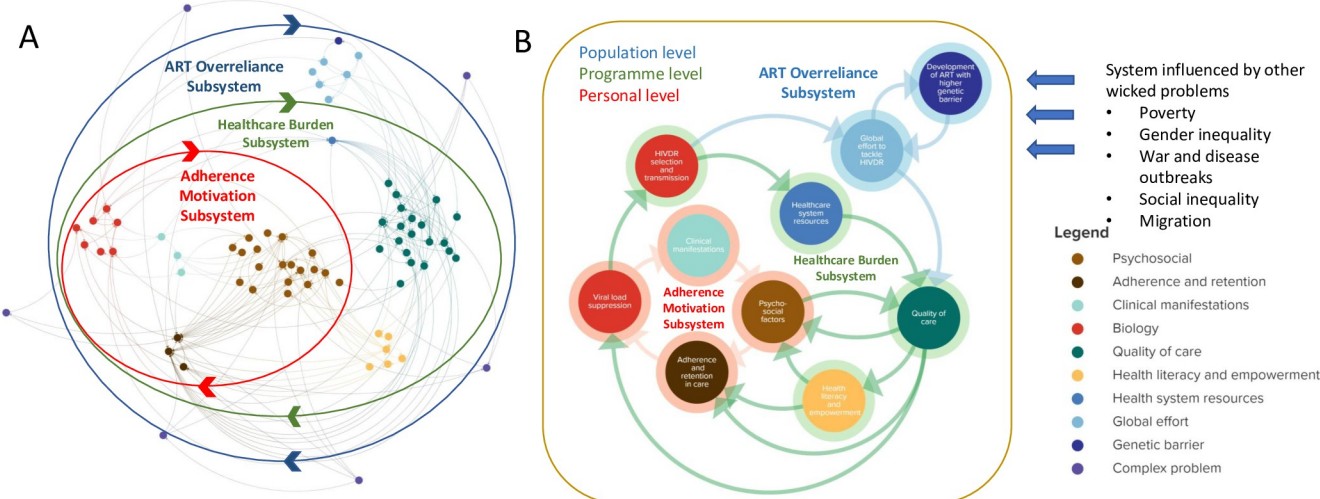

**Fig 6. Summarizing a complex system.** A) Detailed system of factors influencing HIVDR. The main feedback loops or subsystems are highlighted with colored circles. B) The same system, condensed into the main feedback loops and with the main exogenous factors represented on the outside. Each cluster in panel A is represented as a single element in panel B, represented with the same color in the core of the element. All connections between two clusters in panel A are represented as one connection in panel B. This way, the main dynamics of the system are represented in a more condensed and comprehensible format. Figure adapted from Kiekens et al. [19] and for illustrative purposes only.

highest in-degree was "adherence", which is not surprising because adherence is a well-studied factor with a known correlation to HIVDR and many influencing variables [29, 30]. The out-degree describes the number of outgoing connections (the number of elements thought to be influenced by the element of interest). In our case study, "understanding of HIV infection and treatment" had the highest out-degree, indicating that this element is perceived to exert the most influence on the rest of the system. Elements with a high out-degree but a low in-degree might be good candidates for leverage points in a system as they impact several parts of the system but are not influenced by many other elements. For example, social support is influenced by the status disclosure of the patient and the knowledge the family members have about HIV, while it has a direct impact on five different factors in the system, such as acceptance of HIV status and help with adherence. Another quantitative method by Finegood et al. can be used to quantitatively compare two systems maps visualizing the same system but from different points of view [31]. In the Finegood method, elements (based on the same thematic coding for both maps) are divided into clusters and inter- and intra-cluster relationships are compared. However, an important note has to be made concerning both methods. When interpreting these metrics, one needs to consider the coding choices made earlier in the process. Coming back to the example used before, psychological wellbeing can have an in-degree of four as it is influenced by stigmatisation, self-stigmatisation, discrimination and gossip, or it can have an in-degree of one if the researcher has decided to group all four elements in one. In both cases, the content of the map is the same, but the in-degree metric will be different (Fig 5). The authors therefore advise to be cautious when using such metrics as a supportive tool during analysis and always ground findings in qualitative evidence. Other methods to quantify causal loop diagrams and to select desirable future scenarios have been described in the literature [11, 32].

After analysis, it is advised to link back to the stakeholders and population in order to validate the findings. This is an important step in order to verify whether data was correctly interpreted and whether no major elements were overlooked. This can be done during participation in a conference if the target population are experts, or through a workshop, peer debriefings or member checks.

**4. Exploring the system dynamics.** While a systems map is a static representation of a CAS, in reality systems are constantly evolving and reorganizing when changes occur. Uncovering the potential for adaptiveness in a CAS requires an understanding of what is contributing to such dynamics. Once the system is mapped, one therefore needs to explore the characteristics that have the potential to lead to adaptiveness [5, 6]. These characteristics may be difficult to represent in a static systems map, which makes it all the more important to study them separately. In Table 3 we illustrate some of these characteristics with an example of our case study.

**5. Continuous transdisciplinary reflexivity.** Though linearly described above, the process of data collection, mapping and analysis is actually an iterative process in which more data is collected based on newly gained insights and different mapping and analysis rounds are needed to explore different lines of thinking. Throughout all this, it is important that the researchers adopt a transdisciplinary approach, truly integrating the knowledge of different disciplines while transcending disciplinary boundaries. As our education system today is largely disciplinary, a quantitatively trained researcher will have to immerse him- or herself into the qualitative research paradigm and vice versa. Posner. et al. and McGregor describe this transition from mono- to transdisciplinarity as a conceptual shift in ones ideas about reality, logic and knowledge [33, 34]. Throughout the systems mapping and analysis process the researchers needs to be constantly aware of their potential disciplinary bias and need to search for active ways to avoid this, such as seeking continuous feedback from other disciplines or stakeholders, organizing group validation sessions etc. Moreover, the researcher should be aware that the systems map will never be truly finished as situations and conditions are continuously changing. Rather, the map should be seen as a dynamic tool that serves the research purposes, while staying open for changes. In short, we advise researchers to 1) immerse themselves into the literature and research paradigm of other relevant disciplines before starting the research, 2) aim for multi-disciplinarity within the research team, 3) continuously reflect on

**Table 3. Dynamic characteristics of CASs.**

| Characteristic | Explanation | Example |
|---|---|---|
| Emergence | Spontaneous behaviour which arises when individual actors or elements reorganize themselves into a bigger whole. | In order to prevent HIVDR, it is important that PLHIV take their medication on a daily basis. When there is a stock-out, healthcare workers organize themselves in WhatsApp groups in order to re-divide the stock and provide all patients with their doses. |
| Path dependence | Events that started in the same point, can lead to different outcomes, depending on the choices that are made during the process. | When a patient discloses their HIV status to family members it can lead to an increased social support and a better adherence, but also to stigmatisation, a decreased self-image or for example loss of employment opportunities. |
| Feedback loop | A series of elements that influence each other in a circular motion. | PLHIV need to take their medication with a meal in order to avoid side effects. When medication is taken daily, the patient will feel healthy and will be able to work and have access to daily meals as well as provide for their family. This reinforcing feedback loop is also used by healthcare workers to motivate PLHIV. |
| Tipping point | A point at which the system will rapidly change and eventually settle into a new balanced state. | Stigmatisation of PLHIV is for a large part caused by a lack of information and knowledge on the nature of the infection and the transmission modalities. When the point is reached where enough people have sufficient knowledge, and community stigmatisation decreases, it is possible that the system (which is now strongly influenced by stigmatisation), will rapidly adapt into a new state. |
| Culture | The shared values and believes which are intrinsically part of the system and which, as such, contribute to the system dynamics and information flows. | In the Tanzanian culture, religious leaders and traditional healers play a prominent role. PLHIV may believe they are punished by god when they first find out about their status, or believe they will get cured by praying. Religious leaders and traditional healers may therefore play an important role in the spread of correct information and the motivation to adhere to the medication. |

Elements that contribute to the dynamics of a CAS, illustrated with an example of our case study.

the possibility of disciplinary bias, and find ways to minimise it and 4) accept the dynamic and unfinished nature of systems maps.

## Conclusion

Systems approaches are increasingly used to study complex health problems. The development of a systems map of the factors influencing the topic under investigation is not only useful as a process of transcending disciplinary boundaries and creating a holistic overview of the situation, but also as a means of gaining deep systems related insights in the underlying dynamics that drive this issue. In this article we have laid out a practical guideline for developing and analyzing a systems maps for complex public health issues. Such systems maps can be used to identify the root causes and intervention points in the system and to understand the dynamics that lead to the adaptiveness of a system. They may also potentially serve as a basis for further quantitative modelling.

## Supporting information

**S1 File. Interview guide international experts.** Interview guide for the international expert interviews. This interview data is here used to illustrate our methodology and has been published elsewhere [19].
(PDF)

**S2 File. Interview guide PLHIV.** Interview guide for the interviews with PLHIV in Dar es Salaam, Tanzania. This interview data is here used to illustrate our methodology and has been published elsewhere [20].
(PDF)

**S3 File. Interview guide local actors.** Interview guide for the local actors in Dar es Salaam, Tanzania. This interview data is here used to illustrate our methodology and has been published elsewhere [20].
(PDF)

**S4 File. Codebook.** Codebook used for the systems maps represented in Figs 4 and 6. For privacy reasons quotes and references to interviews are not included. Tab 1 contains a weblink to an interactive version of the systems maps. Tab 2 represents the elements, their definition and a categorisation into layers (for Fig 4A), subsystems (for Fig 6B) and clusters (for Figs 4B and 6A). Tab 3 contains the connections, their polarity and their definitions. Tab 4 and 5 contain the Kumu code for both versions of the systems maps.
(XLSX)

## Acknowledgments

The authors would like to thank Dr Catherine Decouttere and Dr Idda Mosha for sharing their methodological expertise with us.

## Author Contributions

**Conceptualization:** Anneleen Kiekens, Bernadette Dierckx de Casterlé, Anne-Mieke Vandamme.

**Methodology:** Anneleen Kiekens, Bernadette Dierckx de Casterlé, Anne-Mieke Vandamme.

**Supervision:** Anne-Mieke Vandamme.

**Visualization:** Anneleen Kiekens.

**Writing – original draft:** Anneleen Kiekens.

**Writing – review & editing:** Anneleen Kiekens, Bernadette Dierckx de Casterlé, Anne-Mieke
   Vandamme.

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
