## [Decision Letter · Decision Letter 0]

1 Nov 2021

PONE-D-21-29069Qualitative systems mapping for complex public health problems: a practical guidePLOS ONE

Dear Dr. Kiekens,

Thank you for submitting your manuscript to PLOS ONE. After careful consideration, we feel that it has merit but does not fully meet PLOS ONE’s publication criteria as it currently stands. Therefore, we invite you to submit a revised version of the manuscript that addresses the points raised during the review process.

Both reviews are quite detailed and contain a number of comments, some of particular relevance for a positive decision about publication, that need to be addressed explicitly and in full. In particular, while both Reviewers expressed the opinion that the work could potentially represent an important scientific contribution, at the same time both lamented insufficient quality in the presentation of results, of the robustness of conclusions, and in other aspects that should have been treated with greater care. 

Some doubts about the ethic statement and the availability of data should be clarified. For the question regarding the possible need of an ethical approval for the experiment, please note that PLOS ONE specifies that " This (Ethic) statement is required if the study involved: • Human participants", and your study seems to involve human participants. About data availability, PLOS ONE requires it in any case, with only few exceptions when legal obligations forbid the disclosure.Figure 1 is missing. This is obviously a distraction error, but be advised that, the next time, an incomplete submission will automatically lead to a rejection from desk.

For the resubmission, it is expected that particular care will be put in thoroughly address all reviewers comments, by improving the manuscript accordingly and providing detailed answers. Changes should be clearly highlighted in the mark-up copy.

We look forward to receiving your revised manuscript.

Kind regards,

Marco Cremonini, Ph.D.

University of Milan

Academic Editor

PLOS ONE

Journal Requirements:

2. When reporting the results of qualitative research, we suggest consulting the COREQ guidelines  or other relevant checklists listed by the Equator Network, such as the SRQR, to ensure complete reporting (http://journals.plos.org/plosone/s/submission-guidelines#loc-qualitative-research). Moreover, please provide the interview guide used as a Supplementary File." Do not ping with follow up, thanks!

3.PLOS ONE does not permit references to unpublished data; therefore, we request that you either include the referenced data or remove the instances of "data not shown," "unpublished results," or similar.

4. We note that Figures 3 and 5 in your submission contain copyrighted images. All PLOS content is published under the Creative Commons Attribution License (CC BY 4.0), which means that the manuscript, images, and Supporting Information files will be freely available online, and any third party is permitted to access, download, copy, distribute, and use these materials in any way, even commercially, with proper attribution. For more information, see our copyright guidelines: http://journals.plos.org/plosone/s/licenses-and-copyright.

1. You may seek permission from the original copyright holder of Figures 3 and 5 to publish the content specifically under the CC BY 4.0 license. 

Reviewers' comments:

Reviewer's Responses to Questions

**Comments to the Author**

1. Is the manuscript technically sound, and do the data support the conclusions?

Reviewer #1: Partly

Reviewer #2: Partly

2. Has the statistical analysis been performed appropriately and rigorously? 

Reviewer #1: N/A

Reviewer #2: N/A

3. Have the authors made all data underlying the findings in their manuscript fully available?

Reviewer #1: No

Reviewer #2: Yes

4. Is the manuscript presented in an intelligible fashion and written in standard English?

Reviewer #1: Yes

Reviewer #2: Yes

5. Review Comments to the Author

Reviewer #1: This was a tricky decision to make. On the one hand, the manuscript represents a potentially important contribution to the scientific literature on CAS. The writing style and line of argument was sound throughout, and I found the practical suggestions for mapping out a CAS to be helpful. This would be the case for any researcher working in a health-related capacity and studying processes of change, intervention effectiveness, etc. The specific example is pretty interesting too: Even the 'simplest' level here - the interaction of drugs, virus, and patient physiology, is clearly a CAS before we even add in the psychological, social, political and economic processes. It is for these reasons I'm suggesting a major revision, but I had several concerns, some quite serious about the manuscript which would need addressing before I could recommend publication:

1. The statement on ethics has been left blank, but the data underlying the manuscript, as far as I can tell, came from interviews with human participants about a sensitive topic. This should be addressed in full, including details of institutional approval etc.

2. This journal requires authors to make datasets publicly available, but I could not see any dataset attached to the submission. It looks like this might have been an oversight, but please be sure to include this with the re-submission.

I had a few substantive methodological points too:

1. The approach described is based on interview data, but the manuscript lacks detail on this - how many participants were recruited, were they all PWHIV or did you include other people working at other parts of the system (eg policy makers, health professionals, etc?)

2. On a related point, do you think interviews alone is enough to understand the dynamics of a system, or ideally should researchers include things like participant observation, documentary and policy analysis? I think at the very least you'd need interviews with a decent cross-section of the types of agent who interact within the system. This could be a useful part of any guide for developing CAS visualisations.

3. Although a lot of effort has clearly been put into coding, and explaining some of the process leading to drug resistance, some of the figures didn't make a great deal of sense to me. For example, what do the 'nodes' in fig 3 actually represent? This looks like some sort of social network map, but what are the individuals and how were they derived? What are the 'overarching factors' for instance? Figure 5A suffers from the same sort of problem: the nodes have general labels like 'psychosocial', but it's not clear exactly what psychosocial factors/ processes are at work in the system, nor how their relationships with the other nested levels of the CAS have been derived.

These visualisations also seem at odds with the qualitative approach taken, as they look like quantitative visualisations of social networks which I have seen in SNA articles, and which I've recently used myself with a colleague. Diagrams that show directions of causality between specific, more precisely defined actors and factors would work better in my view.

4. Where is Figure 1?

Be sure to review for typos, eg 'evens that started in the same point' in Table 2.

Reviewer #2: Abstract

The abstract describes the paper’s goals clearly and immediately sheds light on the adopted approach: proposing system maps to analyze a complex health issue, understanding the root causes, and intervening. The authors also specify that they based the paper on the insight from a previous study of theirs.

Data sources

The Authors specified in the Plos format that all relevant data are within the manuscript and its supporting information file (SI)

Introduction

The authors spend words to connect with the existent literature. They highlight that it is mainly theoretical, then they move to illustrate the concepts of system mapping. They provide adequate and accurate references to support their statements.

They also provide a concise introduction of system mapping through causal loop diagram notation. The authors could consider that it may confuse non-expert readers and evaluate whether distinct CLD by other methods (pictures? Schema? Table?). For readers’ learning, the Authors could also consider citing books about this method and more in general about CAS. (Only as examples, John Sterman’s contributions).

The introduction ends by clearly declaring the manuscript goals: Provide practical guidelines for using system mapping to investigate rich data on public health challenges. They specify that they demonstrate their guidelines based upon a case study.

The verb “demonstrate” has a strong meaning. Did the Authors use it on purpose?

Case Study

The case study description is clear, but I could not find Fig.1.

Data Collection

The Authors specify that they have collected data through individual, semi-structured interviews. The authors adduced suitable reasons for choosing this approach, but they should better specify the reference literature that supported their choice.

The Authors should also better specify:

• Why starting with an open question could/should/can reduce bias

• What they intend with “believes” and what is the literature they had referred to

The Authors should also discuss how they have mitigated the risk of hidden biases that can emerge in sequential questions caused by selective reinforcing loops. As for any ethnographic approach, bias existence should be taken for granted but explored carefully.

Mapping a complex adaptive system

The authors state that they adopted the QUAGOL method and correctly quoted the source. The following description of their application of quagol is clear and sounds correct. Table 1 aims to give an example, but it is unclear how the reader should use it. The Authors should consider clarifying how it relates to mapping complex adaptive systems in the main document, not in the table caption.

The Authors should also circumscribe the concept of mental models. While they are probably aware of the diverse interpretations, showing the connections between reasoning, making sense, deciding, and acting could be beneficial for the less expert readers – which seems to be their target, perhaps (works of Gerd Gigerenzer, Gary Klein, John Sterman, Senge et al, …).

That is of paramount importance as their method relies to some extent on the acceptance of these – here implicit – connections.

The Authors explore their case study in light of their approach thoroughly and accurately. Figures support their discourse, but they could consider adding either a schema or flow diagram to help the reader follow their examination.

The examination of the “depth of the system”, supported by figure 4, ends in an ambiguous description: are points 1) and 2) either rules or norms to follow? Or are they only suggestions the authors have made for increasing the robustness of this approach?

Could a less talkative and more schematic approach help? The whole section is informative, but the authors should carefully consider revising it, splitting the description of the guidelines from collateral enrichments and comments (beneficial, indeed!).

Transdisciplinary and system mapping

The section is informative, but the Authors should consider delivering sharper statements. While the considerations are undoubtedly correct and impactful, the less expert readers could not figure out how to use them.

Conclusions

The paper explores a relevant subject, and the Authors describe all the connections with previous literature at a sufficient level. They clearly express the paper's goals and the gaps it contributes to filling.

Reviewer's Syntesis

The paper explores a relevant subject, and the Authors describe all the connections with previous literature at a sufficient level. They clearly express the paper's goals and the gaps it contributes to filling.

The text is attractive and easy to read. Examples are clear, but sometimes they do not support the concept explanation sharply, while the readers can find themselves wondering about diverse interpretations.

The papers do not state the announced guidelines explicitly, while they are sparse and sound more like suggestions and reflection hints than normative guidelines.

Though the actual limits, this paper is grounded in peer reviews studies, and its goals, approach, and content are impactful and relevant. The way it transmits the subject is sometimes incoherent with its goals. While it promises guidelines, it often delivers informative and accurate reflections. Though they are helpful for those who want to adopt CAS perspectives and apply causal loop diagrams, the readers risk ending up in landscape plenty of theories without a clear map.

6. PLOS authors have the option to publish the peer review history of their article (what does this mean?). If published, this will include your full peer review and any attached files.

Reviewer #1: **Yes: **Tim Gomersall

Reviewer #2: **Yes: **Andrea Montefusco

---

## [Author Response · Author response to Decision Letter 0]

29 Dec 2021

Dear reviewers,

Thank you for taking the time to carefully assess our submission. We have reviewed our manuscript accordingly and have addressed all your points raised.

Below you can find a detailed reply to each of your remarks. Please note that the indicated page and line numbers refer to the version of the manuscript containing track changes.

Reviewer #1: This was a tricky decision to make. On the one hand, the manuscript represents a potentially important contribution to the scientific literature on CAS. The writing style and line of argument was sound throughout, and I found the practical suggestions for mapping out a CAS to be helpful. This would be the case for any researcher working in a health-related capacity and studying processes of change, intervention effectiveness, etc. The specific example is pretty interesting too: Even the 'simplest' level here - the interaction of drugs, virus, and patient physiology, is clearly a CAS before we even add in the psychological, social, political and economic processes. It is for these reasons I'm suggesting a major revision, but I had several concerns, some quite serious about the manuscript which would need addressing before I could recommend publication:

Reply: Thank you for your appreciation of our manuscript and for recognizing the relevance of having a practical guideline for the mapping and analysing of CAS. We have addressed your concerns below and remain available for further clarifications if needed. 

Reviewer: The statement on ethics has been left blank, but the data underlying the manuscript, as far as I can tell, came from interviews with human participants about a sensitive topic. This should be addressed in full, including details of institutional approval etc.

Reply: The interview data we refer to in this manuscript are used as a case study to illustrate our methodology and has been published elsewhere in two publications:

Kiekens A, Mosha IH, Zlatić L, Bwire GM, Mangara A, Dierckx de Casterlé B, et al. Factors Associated with HIV Drug Resistance in Dar es Salaam, Tanzania: Analysis of a Complex Adaptive System. Pathog 2021, Vol 10, Page 1535. 2021;10: 1535. doi:10.3390/PATHOGENS10121535

Kiekens A, Dierckx de Casterlé B, Pellizzer G, Mosha I, Mosha F, Rinke de Wit T, et al. Identifying mechanisms behind HIV drug resistance in Sub-Saharan Africa: a systems approach. BMC Public Health. : Pre-print under peer review. doi:10.21203/rs.3.rs-764469/v1

We provided the editor with all the necessary information on the ethical approval for those two publications, to be published at the editor’s discretion.

Reviewer: This journal requires authors to make datasets publicly available, but I could not see any dataset attached to the submission. It looks like this might have been an oversight, but please be sure to include this with the re-submission.

Reply: The datasets used to illustrate our methodological guideline has been published in a pre-print version of an article under review in BMC Public Health. We have included the dataset in the supplementary material (S4 File).

Reviewer: I had a few substantive methodological points too:

The approach described is based on interview data, but the manuscript lacks detail on this - how many participants were recruited, were they all PWHIV or did you include other people working at other parts of the system (eg policy makers, health professionals, etc?)

Reply: We have added some more information on the interviews conducted in our case study. We have also clarified that this case study has already been described in two separate publications and provided references. We feel it would be repetitive to provide all the details on the case study in this manuscript. We therefore specified that the case study is here used to illustrate our guidelines with examples. [page 5, line 110-115]

“These systems maps were informed by interviews with 15 international experts, 12 PLHIV in Dar es Salaam and 10 Local actors, who through their daily activities regularly come into contact with PLHIV in the study site. The findings of these studies are described in two separate publications [19,20]. In the next sections we use examples from this case study to illustrate our guide.”

Reviewer: On a related point, do you think interviews alone is enough to understand the dynamics of a system, or ideally should researchers include things like participant observation, documentary and policy analysis? I think at the very least you'd need interviews with a decent cross-section of the types of agent who interact within the system. This could be a useful part of any guide for developing CAS visualisations.

Reply: Indeed, we suggest to carefully recruit interviewees with the aim of covering all possible aspects of the system. Although we haven’t used additional methodology such as observations or document analysis, we do find this a good suggestion and have added it to the text. [page 6-7, line 130-138]

“Participants should be recruited with the aim of obtaining a full picture of all aspects of the system. Next to interviewing patients and healthcare workers, one might therefore also consider interviewing people who are somewhat further removed from the core problem but are still in touch with certain parts of it. For example, architects designing certain hospital area’s relevant for the topic under investigation or religious leaders who provide spiritual support to patients could provide unique insights into the topic. Next to the interviews themselves, other types of data such as participant observation and document analysis could also be used to triangulate the data and increase the validity of the results.”

Reviewer: Although a lot of effort has clearly been put into coding, and explaining some of the process leading to drug resistance, some of the figures didn't make a great deal of sense to me. For example, what do the 'nodes' in fig 3 actually represent? This looks like some sort of social network map, but what are the individuals and how were they derived? What are the 'overarching factors' for instance? Figure 5A suffers from the same sort of problem: the nodes have general labels like 'psychosocial', but it's not clear exactly what psychosocial factors/ processes are at work in the system, nor how their relationships with the other nested levels of the CAS have been derived.

These visualisations also seem at odds with the qualitative approach taken, as they look like quantitative visualisations of social networks which I have seen in SNA articles, and which I've recently used myself with a colleague. Diagrams that show directions of causality between specific, more precisely defined actors and factors would work better in my view.

Reply: Figure 3 (now figure 4) and figure 5a (now figure 6a) represent an overview of all elements influencing HIVDR as mentioned in the interviews and the connections between those elements. It does not represent interactions between people (as is the case in a social network map) but between factors. Moreover, both included systems are a qualitative representation of the system. We agree that it is difficult to understand the figures without further information. Therefore, we added a link to an online interactive version of both systems maps where the reader can browse through the systems maps, zoom in and click on elements and connections to gain a better understanding of how what the maps represent. We have added this to the text and we have included an overview of all the elements and connections present in this figure as supplementary file 4. The aim of figure 4 is to show that different representations of a same system are possible. The aim of figure 6 is to show how to summarize a system. For a detailed explanation and interpretation of the systems maps we refer the related publication:

Kiekens A, Dierckx de Casterlé B, Pellizzer G, Mosha I, Mosha F, Rinke de Wit T, et al. Identifying mechanisms behind HIV drug resistance in Sub-Saharan Africa: a systems approach. BMC Public Health. : Pre-print under peer review. doi:10.21203/rs.3.rs-764469/v1

“In both visualizations each element is a factor influencing HIVDR as mentioned in the interviews and each connection indicated the relationship between those factors. An overview of all elements and connections is included in these maps as well as an interactive version of the systems maps where the reader can zoom in and click on elements and connections for more information is included in the supplementary files (S4 File) [27].” [page 13-14 line 231-236]

Reviewer: Where is Figure 1?

Reply: We confirm that we did submit Figure 1 as we can see it in the file inventory of the editorial manager. We are not sure what went wrong but we will make sure to submit Figure 1 again with the resubmission. Note that after revision, this figure is now figure 2.

Reviewer: Be sure to review for typos, eg 'evens that started in the same point' in Table 2.

Reply: We reviewed the text for spelling and grammar errors. 

Reviewer #2: Abstract

The abstract describes the paper’s goals clearly and immediately sheds light on the adopted approach: proposing system maps to analyze a complex health issue, understanding the root causes, and intervening. The authors also specify that they based the paper on the insight from a previous study of theirs.

Data sources

The Authors specified in the Plos format that all relevant data are within the manuscript and its supporting information file (SI)

Introduction

The authors spend words to connect with the existent literature. They highlight that it is mainly theoretical, then they move to illustrate the concepts of system mapping. They provide adequate and accurate references to support their statements.

They also provide a concise introduction of system mapping through causal loop diagram notation. 

The authors could consider that it may confuse non-expert readers and evaluate whether distinct CLD by other methods (pictures? Schema? Table?). For readers’ learning, the Authors could also consider citing books about this method and more in general about CAS. (Only as examples, John Sterman’s contributions).

Reply: Thank you for your appreciation of our abstract and introduction. As suggested, we have added a visual example to understand causal loop diagrams (Fig. 1, page 3 and text page 3, line 58-62). We opted to only include this example in order not to distract the reader with too much detail on the different types of mapping, as this manuscript particularly focusses on causal loop diagrams. We have also added a reference to Sterman’s book to provide the reader with some background reading material. [page 4, line 74-75].

“An often-used example of a causal loop diagram is that of the heating and the thermostat (Fig 1). When the room temperature drops to a certain point, the thermostat will automatically increase the heating (negative causality). When the heating is on, the room temperature will increase (positive causality) up until the point in which the desired temperature is reached. 

Figure 1: causal loop diagram example. Thermostat room temperature regulation as a (simplified) example of a causal loop diagram. 

While in this manuscript we primarily focus on the mapping of causal loop diagrams, there are also other types of mapping such as stock and flow diagrams, which is a more quantitative approach to systems mapping and can be used to study the dynamic behaviour of a system over time, or social network analysis which aims at visualising and studying the relationships between social actors [10,11].”

“Systems dynamics and types of modelling have been thoroughly described by Sterman (2000) [12].”

Reviewer: The introduction ends by clearly declaring the manuscript goals: Provide practical guidelines for using system mapping to investigate rich data on public health challenges. They specify that they demonstrate their guidelines based upon a case study. The verb “demonstrate” has a strong meaning. Did the Authors use it on purpose?

Reply: We understand that the verb “demonstrate” can have a quantitative connotation. As we meant that we use a case study to provide examples for our guide, we rephrased the sentence. [page 4-5, lines 93 – 96]

“We use a case study concerning HIV drug resistance in Sub-Saharan Africa as a practical example to clarify how complex data can systematically be collected, mapped and analysed while incorporating the principles of transdisciplinarity every step of the way (Fig 2).”

Reviewer: Case Study

The case study description is clear, but I could not find Fig.1.

Reply: We confirm that we did submit Figure 1 as we can see it in the file inventory of the editorial manager. We are not sure what went wrong but we will make sure to submit Figure 1 again with the resubmission. Note that after revision, this figure is now figure 2.

Reviewer: Data Collection

The Authors specify that they have collected data through individual, semi-structured interviews. The authors adduced suitable reasons for choosing this approach, but they should better specify the reference literature that supported their choice.

Reply: We have added some references to literature about interviewing about sensitive topics and one reference which describes that individual interviews, in contrast to group sessions, allows the researcher to delve deep into personal and social matters. [page 4, line 83-90]

“ For example, the topic under investigation might be highly stigmatised in the community, therefore not allowing participants to speak freely during a group model building session, the participants perspectives on the topic may be too diverse to organize a common discussion (e.g. technical experts vs family and friends), or participants might live in different parts of the world, making a physical meeting organizationally challenging [15,16]. In some cases, individual interviews are also preferred when one aims to understand the individual mental models of stakeholders separately before generating an integrated overview of the combined viewpoints [17].”

Reviewer: The Authors should also better specify:

• Why starting with an open question could/should/can reduce bias

• What they intend with “believes” and what is the literature they had referred to

The Authors should also discuss how they have mitigated the risk of hidden biases that can emerge in sequential questions caused by selective reinforcing loops. As for any ethnographic approach, bias existence should be taken for granted but explored carefully.

Reply: Indeed, the interview analysis is an interpretative process and bias should be mitigated as much as possible. We started with a broad and open question about the complex problem in general, and not about only one aspect of it, because we wanted to allow the interviewee to share what is most important to them first. This way we avoided introducing bias ourselves by asking a question about a specific part of the system. With these open questions, we want to capture rich and nuanced data. To further avoid bias, especially for our case study example in which we interviewed PLHIV, we took some measures to create trust between the interviewer and the interviewee. For example, the interviews were done by a local researcher in the participant’s native language. There were no other people in the room and the interviews were held on a location which could not be associated with HIV, and therefore stigmatized. With regards to the potential bias with the sequential questions, we would also like to clarify that the chain of causality was built up several times throughout one interview, each time starting from a broad question. This way, some topics were covered from different angles, which also helped us to better understand how the participant experienced the situation and whether there may be bias. Furthermore, the questions were not cognitive questions but aimed to understand situations in the daily life of the interviewees which also lowers the chance of the interviewee feeling the need to give certain “correct” answers or hide their thoughts. We also added an example to explain that with “believes” we mean the underlying assumptions the participant has about the topic. All of this is added in the text on page 7-8, line 154-175.

“In order not to bias the data collection towards certain assumptions the researcher may have, it is advised to start the interview with a broad, open question about the complex problem, rather than asking a question about a single aspect of the problem. This way, the participants are inclined to start by expressing the aspect of the complex problem most important to them and the interviewer can explore the main believes and experiences the participant has to share about the topic. For example, an initial question like “how do you experience being HIV positive” may reveal to the interviewer that the patient’s whole perception of his or her HIV infection is based on the believe that it is a punishment of God. This information is important for the interpretation of the rest of the interview and may not have come up if the interview had started with a focus on a certain aspect of the system, such as the question “how do you perceive the healthcare service you receive?”. After this first question, the interviewer may continue covering a list of specific topics, retrieved from the literature or which came up in previous interviews. When a question is answered by “A happens because of B” the interviewer can ask for specific examples or experiences that support this claim and subsequently delve deeper into other possible underlying causes aiming to obtain the structure “A happens because of B, which is caused by C, D and E, etc.” This continues until a sufficient level of depth is reached or until the insights of the interviewee are exhausted, at which point the chain of causality may be built up further during interviews with other participants. Such chain of causality is built several times within one interview, each time starting from an open question. To further reduce bias, the interview circumstances should be well thought-through in order to create trust between the interviewer and interviewee. For example, for interviews with PLHIV are best done in their native language and in a location that cannot be perceived as stigmatizing.”

Reviewer: Mapping a complex adaptive system

The authors state that they adopted the QUAGOL method and correctly quoted the source. The following description of their application of quagol is clear and sounds correct. Table 1 aims to give an example, but it is unclear how the reader should use it. The Authors should consider clarifying how it relates to mapping complex adaptive systems in the main document, not in the table caption.

Reply: We have added an explanation on how to use the table in the text, as suggested. [Page 9, line 199-201]. Note that after revision table 1 is now table 2.

“In Table 2 we explain the types of data that can be stored behind one element, using the element “accessibility of healthcare centre” as an example.”

Reviewer: The Authors should also circumscribe the concept of mental models. While they are probably aware of the diverse interpretations, showing the connections between reasoning, making sense, deciding, and acting could be beneficial for the less expert readers – which seems to be their target, perhaps (works of Gerd Gigerenzer, Gary Klein, John Sterman, Senge et al, …).

That is of paramount importance as their method relies to some extent on the acceptance of these – here implicit – connections.

Reply: Thank you for pointing us to some reference literature. We have provided an explanation on what mental models are and why they are important for modelling a complex adaptive system. [page 13, line 211-215]

“Mental models are graphical representations of how people internally understand causal relationships between elements to make sense of a complex problem [18,19]. They often unconsciously affect our behavior or decision maker and are useful for us to gain a deeper understanding of the interviewees way of thinking about the problem [20].” 

Reviewer: The Authors explore their case study in light of their approach thoroughly and accurately. Figures support their discourse, but they could consider adding either a schema or flow diagram to help the reader follow their examination.

Reply: Thank you for noticing this. In fact, Figure 2 contains diagram with an overview of the suggested method. We are aware that this figure did not reach you before and we hope this issue will be resolved during resubmission.

Reviewer: The examination of the “depth of the system”, supported by figure 4, ends in an ambiguous description: are points 1) and 2) either rules or norms to follow? Or are they only suggestions the authors have made for increasing the robustness of this approach?

Reply: Indeed, we suggest to apply these two questions to the whole systems map in order to have a common criterion to use for all elements and connections end, indeed, increase the robustness. However, we think other criteria (possibly topic-related) could also be used for this. This is why we do not claim this to be a rule that must be followed by anyone developing a systems map, but we rather include it as a suggestion. We have clarified this in the text. [page 16, line 287-297]

“In the rest of this paragraph we suggest some strategies for the simplification of systems maps. Other strategies (possibly topic dependent) could also be used. More important is to consequently apply the strategy to the whole systems map. When in doubt whether two elements should be merged or not, we suggest the researcher asks two questions: 1) are there significant differences in nuance between the content of both elements? And 2) do both elements have different connections to other elements? If the answer to both questions is “No”, the elements can be merged into one.”

Reviewer: Could a less talkative and more schematic approach help? The whole section is informative, but the authors should carefully consider revising it, splitting the description of the guidelines from collateral enrichments and comments (beneficial, indeed!).

Reply: We understand it is difficult for the reader to differentiate between the different steps of the guideline and our examples. We have tried to resolve this by giving titles to the sections, so that the article takes the shape of a step by step guide. There are four main sections which can also be found back in figure 2, and several subsections.

1. Data collection

1.1. Choosing a data collection method and participant selection

1.2. Preparing and conducting semi-structured interviews

2. Data analysis and mapping

2.1. Interview analysis

2.2. Coding

2.3. From codes to systems map

2.4. Setting systems boundaries

2.5. Determining the depth of the system

2.6. Simplifying the system

3. Analysing the system

4. Exploring the system dynamics

5. Continuous transdisciplinary reflexivity

Moreover, we have reformulated some sections to differentiate better between guidelines and examples from our case study. For example, the section on choosing a data collection method and participant selection on page 6 lines 126-138 and the section on preparing and conducting semi-structured interviews on page 7 line 151-153.

“The first consideration to make is which way of collecting data is most suitable for the topic under investigation. As already explained in the introduction, there are different reasons (both methodological and practical) to opt for either group modelling sessions or individual interviews. This guide focusses specifically on the mapping and analysing of complex data collected by semi-structured interviews. Participants should be recruited with the aim of obtaining a full picture of all aspects of the system. Next to interviewing patients and healthcare workers, one might therefore also consider interviewing people who are somewhat further removed from the core problem but are still in touch with certain parts of it. For example, architects designing certain hospital area’s relevant for the topic under investigation or religious leaders who provide spiritual support to patients could provide unique insights into the topic. Next to the interviews themselves, other types of data such as participant observation and document analysis could also be used to triangulate the data and increase the validity of the results.”

“A semi-structured interview guide should be developed based on the available scientific literature or already existing and validated guides on the topic, and adapted throughout the data collection process when new insights are developed.”

Reviewer: Transdisciplinary and system mapping

The section is informative, but the Authors should consider delivering sharper statements. While the considerations are undoubtedly correct and impactful, the less expert readers could not figure out how to use them.

Reply: We have added some concrete recommendations to this section. [page 22, line 379-383]

“In short, we advise researchers to 1) immerse themselves into the literature and research paradigm of other relevant disciplines before starting the research, 2) aim for multi-disciplinarity within research team, 3) continuously reflect on the possibility of disciplinary bias, and find ways to minimise it and 4) accept the dynamic and unfinished nature of systems maps.”

Reviewer: Conclusions

The paper explores a relevant subject, and the Authors describe all the connections with previous literature at a sufficient level. They clearly express the paper's goals and the gaps it contributes to filling.

Reviewer's Syntesis

The paper explores a relevant subject, and the Authors describe all the connections with previous literature at a sufficient level. They clearly express the paper's goals and the gaps it contributes to filling.

The text is attractive and easy to read. Examples are clear, but sometimes they do not support the concept explanation sharply, while the readers can find themselves wondering about diverse interpretations.

The papers do not state the announced guidelines explicitly, while they are sparse and sound more like suggestions and reflection hints than normative guidelines.

Though the actual limits, this paper is grounded in peer reviews studies, and its goals, approach, and content are impactful and relevant. The way it transmits the subject is sometimes incoherent with its goals. While it promises guidelines, it often delivers informative and accurate reflections. Though they are helpful for those who want to adopt CAS perspectives and apply causal loop diagrams, the readers risk ending up in landscape plenty of theories without a clear map.

Reply: Thank you for your appreciation of our subject and scientific intentions with this manuscript. We have aimed to provide a clearer step by step guide by adding section titles to the text and providing a table with an overview of the steps and their timing within the process [table 1, page 6]. We have also adapted some parts of the text to better differentiate between the guidelines and the examples meant to illustrate the guidelines. We hope that these changes and the ones described above, contribute to clearer formulation of our guidelines. While we have on some places also formulated our statements sharper as requested, we also want the reader to be aware that our guideline should always be considered within the research setting and may therefore be adapted to better suit the conditions. 

We hope these changes have sufficiently improved our manuscript. We are happy to respond to any further questions and comments you may have.

Sincerely,

Anneleen Kiekens, 

corresponding author

---

## [Decision Letter · Decision Letter 1]

11 Feb 2022

Qualitative systems mapping for complex public health problems: a practical guide

PONE-D-21-29069R1

Dear Dr. Kiekens,

We’re pleased to inform you that your manuscript has been judged scientifically suitable for publication and will be formally accepted for publication once it meets all outstanding technical requirements.

Kind regards,

Marco Cremonini, Ph.D.

University of Milan

Academic Editor

PLOS ONE

Additional Editor Comments (optional):

Reviewers' comments:

Reviewer's Responses to Questions

**Comments to the Author**

1. If the authors have adequately addressed your comments raised in a previous round of review and you feel that this manuscript is now acceptable for publication, you may indicate that here to bypass the “Comments to the Author” section, enter your conflict of interest statement in the “Confidential to Editor” section, and submit your "Accept" recommendation.

Reviewer #2: All comments have been addressed

2. Is the manuscript technically sound, and do the data support the conclusions?

Reviewer #2: Yes

3. Has the statistical analysis been performed appropriately and rigorously? 

Reviewer #2: N/A

4. Have the authors made all data underlying the findings in their manuscript fully available?

Reviewer #2: Yes

5. Is the manuscript presented in an intelligible fashion and written in standard English?

Reviewer #2: Yes

6. Review Comments to the Author

Reviewer #2: (No Response)

7. PLOS authors have the option to publish the peer review history of their article (what does this mean?). If published, this will include your full peer review and any attached files.

Reviewer #2: **Yes: **Andrea Montefusco